Otitis media detection using tympanic membrane images with a novel multi-class machine learning algorithm

Alhudhaif Adi 1 a.alhudhaif@psau.edu.sa
http://orcid.org/0000-0001-5256-7648 Cömert Zafer 2
Polat Kemal 3
1 Department of Computer Science, College of Computer Engineering and Sciences in Al-kharj, Prince Sattam bin Abdulaziz University , Alkharj , Saudi Arabia
2 Department of Software Engineering, Samsun University , Samsun , Turkey
3 Department of Electrical and Electronics Engineering, Faculty of Engineering, Bolu Abant Izzet Baysal University , Bolu , Turkey
Gao Zhiwei
Electronic publication date: 2021 Feb 23
Publication date: 2021
Volume: 7
Electronic Location ID: e405
Received 2020 Nov 18; Accepted 2021 Jan 30
Copyright: © 2021 Alhudhaif et al.
Copyright year: 2021
Copyright holder: Alhudhaif et al.
License: This is an open access article distributed under the terms of the Creative Commons Attribution License, which permits unrestricted use, distribution, reproduction and adaptation in any medium and for any purpose provided that it is properly attributed. For attribution, the original author(s), title, publication source (PeerJ Computer Science) and either DOI or URL of the article must be cited.
License URL: https://creativecommons.org/licenses/by/4.0/

Keywords: Biomedical image processing, Decision support system, Otitis media, Convolutional neural networks, Deep learning

Funding: Prince Sattam bin Abdulaziz University, Alkharj, Saudi Arabia This publication was supported by the Deanship of Scientific Research at Prince Sattam bin Abdulaziz University, Alkharj, Saudi Arabia. The funders had no role in study design, data collection and analysis, decision to publish, or preparation of the manuscript.

==============================
Background

Otitis media (OM) is the infection and inflammation of the mucous membrane covering the Eustachian with the airy cavities of the middle ear and temporal bone. OM is also one of the most common ailments. In clinical practice, the diagnosis of OM is carried out by visual inspection of otoscope images. This vulnerable process is subjective and error-prone.

Methods

In this study, a novel computer-aided decision support model based on the convolutional neural network (CNN) has been developed. To improve the generalized ability of the proposed model, a combination of the channel and spatial model (CBAM), residual blocks, and hypercolumn technique is embedded into the proposed model. All experiments were performed on an open-access tympanic membrane dataset that consists of 956 otoscopes images collected into five classes.

Results

The proposed model yielded satisfactory classification achievement. The model ensured an overall accuracy of 98.26%, sensitivity of 97.68%, and specificity of 99.30%. The proposed model produced rather superior results compared to the pre-trained CNNs such as AlexNet, VGG-Nets, GoogLeNet, and ResNets. Consequently, this study points out that the CNN model equipped with the advanced image processing techniques is useful for OM diagnosis. The proposed model may help to field specialists in achieving objective and repeatable results, decreasing misdiagnosis rate, and supporting the decision-making processes.

Introduction

The ear consists of three basic parts which are the outer ear, middle ear, and inner ear. The outer ear forms a pinna and ear canal (Nesterova et al., 2019). The tympanic membrane (TM) separating the outer ear and middle ear is located in the deep of the ear canal. The eardrum has a thin structure that vibrates when sound waves pass through on it. The middle ear is an air-containing space behind the eardrum. The air is transported from the nasal region behind the nose to the middle ear through a small duct called the Eustachian tube (Vanneste & Page, 2019). The task of the Eustachian tube is to equalize the pressure of the air in the middle ear cavity with the atmospheric pressure outside. The vibrations caused by the sound waves in the eardrum are transmitted from the small ossicles (anvil, stirrup, hammer) in the middle ear cavity to the inner ear, and the nerves that transmit the sound to the brain are stimulated. This expresses how the morphological structure of the ear works (Chiong et al., 2020).

Otitis media (OM) or middle-ear infection is a medical term that represents a range of disorders related to the TM (Pichichero, 2013). OM is an inflammation of the middle ear cavity behind the eardrum. It can be observed as a complication starting from the nasal cavity to the upper respiratory tract. The otoscope or the pneumatic otoscope is the basic tools that the Ear Nose Throat (ENT) specialists currently use to examine the TMs (Di Maria et al., 2019). OM is one of the commonly seen health issues, especially in childhood, and may lead to speech defects, facial nerve palsy, hearing loss, and cognitive disorders if it cannot be treated properly and timely (Myburgh et al., 2018). The clinical signs of the OM may include ear pain, ear discharge, headache, undergoing, or recent upper respiratory tract infection, restlessness, and loss of appetite (Albu, Babighian & Trabalzini, 1998).

The common types of OM are acute otitis media (AOM), otitis media with effusion (OME), and chronic suppurative otitis media (CSOM). In addition, although Earwax is not one of the OM types, it can often be described as one of the conditions that cause discomfort. AOM closely related to upper respiratory tract infection (URTI) is inflammation of the middle ear. The signs or symptoms of AOM may reveal with bulging, perforated eardrum, pain, discharge, otalgia, otorrhea, ear fullness (Kim et al., 2020). As a clinical situation, OME is one of the most commonly seen OM disorders that is defined as being of inflammation and collection of fluid in the middle ear cavity without signs or symptoms of acute ear infection (Kaytez et al., 2020). As for CSOM, it is a persistent and insidious ear disorder that discharges into the external ear canal undergoing for over 2 weeks and results in perforation of the TM (Aduda et al., 2013). Earwax includes skin, sweat, hair, and debris. These substances accumulate in the external ear canal, then dry and fall out of the ear. In general, earwax prevents the being infected of TM by dust and other small particles. Though it seems beneficial to ear health, sometimes it may cause discomfort such as pain, ringing in the ear, and decreased hearing (Cömert, 2020, Nakagawa et al., 2018).

When the literature is examined, it can be seen that the related studies based on the computation approaches in the field are relatively limited. An automated algorithm based on the vocabulary and grammar sets, which correspond to the numerical measurements of TMs and the decision rules, respectively, was proposed for determining diagnostic categories of OM. The algorithm was confirmed by otoscopists as well as engineers. The algorithm achieved an 89.9% classification success for normal, AOM, and OME classes (Kuruvilla et al., 2013). A diagnostic model using image processing techniques and decision trees (DTs) was proposed for discriminating five different classes, including the obstructing wax in the external ear canal, normal TMs, AOM, OME, and CSOM OM types. Tailor-made features extraction algorithms were used to describe TM images, and the related features were input to the decision tree. As a result, the model reached an accuracy of 80.60% (Myburgh et al., 2016). A combination of the gray-level co-occurrence matrix (GLCM) and artificial neural network (ANN) was proposed to distinguish normal and acute TMs. The results of the study pointed out that the texture features were found useful to distinguish normal TMs from the acute. The model ensured 76.14% classification achievement (Basaran et al., 2019b). A pretrained deep convolutional neural network (DCNN) called AlexNet was proposed to recognize chronic otitis media (COM) and normal TM samples. The raw otoscope images were input to the AlexNet in this study. To validate the model, 10-fold cross-validation was also used and the model ensured an accuracy of 98.77% (Basaran et al., 2019a). A computational model relying upon the Faster R-CNN and pretrained convolutional neural networks (CNNs) was introduced for separating normal and abnormal TMs. The Faster R-CNN was used to automatically determine the locations of the TMs in otoscope images. This process yielded a patch covering the only TM instead of the full otoscope image. Then, the pretrained CNNs were retrained using a transfer learning approach and using these patches. The model ensured 90.48% classification accuracy with the VGG-16 model (Başaran, Cömert & Çelik, 2020). A multi-class classification task for OM diagnosis was realized using the fuzed fine-tuned deep features in another study. The deep features obtained from the last fully connected layers of the pretrained CNNs were fuzed after the training of the CNNs, and these deep features were applied as the input to the ANN, k-nearest neighbor (kNN), DT, and support vector machine (SVM) classifiers. The most effective results were ensured with an accuracy of 99.47%, sensitivity of 99.35%, and specificity of 99.77% with SVM model (Cömert, 2020). A pediatric otorhynolaryngology diagnosis support system for the visual diagnosis of the ear drum based on the image processing techniques was offered. This study indicates that the compound of TM’s color and auditory canal color significantly improve the model performance (Vertan, Gheorghe & Ionescu, 2011). A large scale otoendoscopic images totally consisting of 10.544 samples were used in another study. An ensemble model including Inception V3 and ResNet-101 was proposed. The success of the model was reported as 93.67% (Cha et al., 2019).

As mentioned above, applying the computerized approaches in OM diagnosis has been found limited. ENT specialists are not fully aware of the benefits of such systems (Goggin, Eikelboom & Atlas, 2007). In many areas, such diagnostic systems are widely used in clinical applications as useful support tools. To overcome subjective evaluation and to ensure more consistent and repeatable results, these systems have been developed and in demand. In this study, we introduce a novel computer-aided decision support system relying on CNN for OM diagnosis. The proposed model exploits the generalization ability of convolution and dense blocks. Moreover, the proposed model is equipped with the channel and spatial model (CBAM) as well as residual block and hypercolumn technique built on attention modules to enhance its classification achievement. The model has ensured a high sensitivity diagnostic model for categorizing different TMs and has yielded promising results.

The rest of the study is organized as follows: the TM datasets and employed methods are summarized in “Materials and Methods”. The experimental results are presented in “Results”. The discussion and conclusion remarks are given in “Discussion” and “Conclusions”, respectively.

Materials and Methods

Eardrum dataset

Up to now, digital acquisition and storage of the TMs have been omitted, whereas this task can be easily fulfilled with limited costs. In this scope, an open-access TM dataset was revealed by the CTG resource group. The dataset consists of a total of 956 raw otoscope images collected from the volunteer patients admitted to Özel Van Akdamar Hospital between 10/2018 and 06/2019 (Cömert, 2020).

The number of normal TM samples is 535 whereas the numbers of AOM, CSOM, Earwax samples are 119, 63 and 140 respectively. By the way, the samples belonging to otitis externa (41), ear ventilation tube (16), foreign bodies in the ear (3), pseudo-membranes (11), and tympanoskleros (28) were collected in the other class. Each otoscope sample in the database was evaluated by three ENT specialists. The samples belonging to different classes were stored in the specified folders named considering OM types. The low-quality images due to lack of light, hand-shake, etc. were also isolated from the database. A visual representation of the TM dataset is given in Fig. 1.

Figure 1 A visual representation of the TM dataset.

Basics of convolutional neural networks

Convolutional neural network ensures an end-to-end learning schema by eliminating the conventional features extraction and feature selection tasks. To this aim, a CNN model frequently includes convolution, pooling and fully connected layers in the architecture (Talo, 2019). The raw data such as signal, image or video is applied to CNN and then the data is passed through deep layers located in the architecture of the network. In this process, one of the basic operations is convolution. A large scale filter family is used in the convolution layers in the CNN to reveal the local as well as global discriminative features. Equation (1) describes the mathematical model of the convolution process. The outputs of the convolution layers are named activation maps (Budak & Güzel, 2020).

(1) Xjl=f(∑i∈Mj⁡Xil−1∗kijl+bjl)

Herein, Mj denotes the input map selection and the activation map of the previous layer is symbolized by Xil−1. This layer is convolved with the learnable kernel kijl and a trainable bias parameter bjl that is used to prevent overfitting. Then, an activation function, f(.) yields the feature map.

The pooling layer is one of the commonly used layers in CNN architectures to decrease the computational cost and to achieve the generalized results. A down-sampling operation is applied in this layer considering average or maximum pooling (Xu et al., 2019). A pooling operation is modeled in Eq. (2).

(2) Xjl=down(Xjl−1)

where the down(.) function corresponds to the down-sampling process, and Xjl−1 is the previous activation.

Fully connected layers ensure a connection between the nodes and activation. These layers including the deep features obtained from the activation maps are used to determine the class of the input data (Başaran, Cömert & Çelik, 2020).

Large-scale datasets are needed to properly train CNNs. Datasets are divided into mini-batches and training is carried out with the help of the optimization algorithms. Determining the most suitable weights for the network is costly and takes a long time. Transfer learning is used to ensure that pre-trained networks such as AlexNet, VGG-Nets, GoogLeNet, and ResNets are adapted to a new area to provide consistent network training for smaller datasets (Altuntaş, Cömert & Kocamaz, 2019).

Optimization algorithms

Stochastic Gradient Descent (SGD) is an iterative method that optimizes the suitability of the function to make it more useful. SGD optimization is a machine learning method that randomly selects some features rather than processing the sample features, calculating their costs in each iteration process. Therefore, instead of calculating a total cost, it contributes to the speed of performance by calculating costs in each iteration process (Ratre, 2019). Restart methods, on the other hand, are preferred for the solution of multi-functional and complex functions, which are frequently used in gradient-free optimizations. The difference of SGD with Warm Restarts (SGDR) method compared to SGD method is to increase the convergence rate in gradients. This has been observed to contribute to the speed and efficiency of the function using the SGDR method. SGDR prefers to reduce the learning rate in multiple functions using semi-cosine curves at the end of each iteration. The process function related to SGDR method is given in Eq. (3). When Eq. (3) is examined; ηt holds the upper and lower limit values of the learning rate of the SGDR optimization method and Tcurrent is the parameter that holds how many iteration passes since the start of optimization (Loshchilov & Hutter, 2017). In addition, in this study, at the end of each epoch, the upper limit value in learning rate was reduced by 20% and the learning rate delay value was taken as 0.8.

(3) ηt=ηmini+12(ηmaxi−ηmini)(1+cos⁡(TcurrentTi)∏)

The Softmax function is an activation function generally used at the end of the model, used in regression and classification processes. In the Softmax processes, the D-dimensional values vector is transferred from the previous layer. Softmax then generates these values in the range [0,1] and produces a reconstructed D-dimensional vector. Sum of probability values in the vector is equal to 1 and the input of the model is transferred to the class with a high probability (Maharjan et al., 2020).

The Adam method is an optimization method that combines the adaptive gradient algorithm (Adagrad) with the advantageous functions of the Root Mean Square Propagation (RMSP) methods. This method is preferred more frequently than other optimization methods and has been observed to give faster results in terms of performance than other optimization methods (Zhao et al., 2019). In the traditional SGD method, it uses a single learning rate value to update all weight parameters and this value does not change during the training period. However, in the Adam method, learning ratios are maintained with each weight update and learning rates are adapted as learning progresses. The Adam optimizes the problem by calculating the exponential moving average of the parameters (Bock & Weiß, 2019). Formulas about Adam are shown in Eqs. (4) and (5). In these equations; Parameters representing (mt) gradient value, V(t) gradient exponential square calculation and β delay rates.

(4) mt=β1mt−1+(1−β1)gt

(5) Vt=β2Vt−1+(1−β2)gt2

Multi-Layer Perceptron (MLP) is a feed forward machine learning method and a controlled learning method that updates the weight parameters used in its algorithm with the back propagation technique. Each layer used in MLP is completely connected with the previous and next layer. It contains input layer, hidden layers and output layer in the MLP structure. The formulas showing the processing steps of the MLP method are shown in Eqs. (6) and (7). In these equations; m are the variables that hold the number of layers, w weight parameter values, α activation layer value and b bias adjustment value. Parameter J represents the difference between the actual output values and the estimated output values. The more this value is minimized, the better the performance of the MLP method (Zhang et al., 2018; Salah & Fourati, 2019).

(6) hw,b(x)=αm

(7) J(W,b;x,y)=12hw,b(x)−y2

In this study, the SGDR parameter is used in the “callbacks” function of the proposed model. Here, the model ensures that the learning rate in each iteration is constantly updated over the previous learning rate values. The use of SGDR is to prevent the proposed model from jamming at the local minimum point. The Adam optimization method is used in the “model.compile” function of the proposed model and is chosen to help update the weight and bias parameters of the proposed model.

Proposed deep learning model

The proposed model is a deep learning model consisting of three basic components. These components consist of attention modules, residual blocks and hypercolumn technique. With the attention modules, it is aimed to increase the performance of the model by attracting attention to the region that should focus on the input images. Thus, time loss of the model is prevented by unnecessary image parts. Layers that make little contribution to the performance of the model have been jump with the Residual blocks, and the features that have been extracted to the afterward layers have been transferred. It is a technique that keeps the features extracted from layers until the model output of each pixel in the input image with hypercolumn and transfers the most efficient feature in it to the output layer. In addition, the model contains convolutional layers, pooling layers and dense layers.

The input size of the proposed model is 224 × 224 pixels. Even if the input data contains variable sizes, it contains the codes that convert the size of the images to the input size. In addition, the proposed model transfers each of the model input images to the convolutional layer by performing one-to-one variation with the augmentation technique (horizontal scroll ratio, vertical scroll ratio, zoom, rotation, sharpness value, etc.). There is absolutely no image duplication process here. There is only one-to-one pretreatment to the existing image. Thus, the extracted features are provided to be more efficient.

The proposed model, which has an end-to-end model feature, has a Softmax activation function that performs the classification process in the last layer. Thanks to this method, the features transferred from the previous dense blocks are processed and probability values are created by Softmax and it transfers the input image to the related class with these probability values. Preferred values related to the augmentation parameters of the model and other important parameters used in the model are given in Table 1. The general structure and design of the proposed model is shown in Fig. 2. The layer parameters and preferred default values in the model design are also shown in Fig. 2. In summary, the proposed model focuses on the related critical areas on the otoscope images using the attention module. The benefits of the abstraction in the model learning process are improved using residual blocks. Also, the most relevant deep features are revealed using the hypercolumn technique in the proposed model.

Table 1 Important parameters and parameter values of the proposed model.

	Optimization	Data augmentations parameters	LR scheduler & parameters	
Parameters	Adam
Beta 1 = 0.9
Beta 2 = 0.999 Decay = 0.0
Initial LR = 0.0001	Vertical Flip = 0.5
Horizontal Flip = 0.5
Random Brightness Contrast = 0.3
Shift Scale Rotate = 0.5
Shift Limit = 0.2
Scale Limit = 0.2
Rotate Limit = 20°	SGDR
Min LR = 1e−6
Max LR = 1e−3
Steps Per Epoch = 5
LR Decay = 0.9
Cycle Length = 10
Mult. Factor = 2	
Environment	Python	
Input size	224 × 224	
Mini-batch size	24	
Framework	Keras	
Loss Type	Categorical Cross-entropy	

Figure 2 The block diagram of the proposed model with the values of the parameters.

Convolutional blocks and dense blocks

The convolutional blocks transmit the feature values/activation maps to the next layer by circulating 3 × 3, 5 × 5 filters on the image as input. In convolutional blocks, there is a direct transfer between layers one after another (Gupta et al., 2019). In residual and dense blocks, this can be transferred after two or three layers.

Dense blocks are the blocks of dense layers that look like residual blocks. The main feature that distinguishes it from residual blocks is that it combines the values obtained at the end of the block instead of collecting them (Zhang et al., 2020). The overall design of these two block structures is shown in Fig. 3. Mini-batch is a parameter that processes inputs in layers simultaneously. This parameter is directly related to hardware features (Nakasima-López et al., 2019). Batch normalization is the parameter that allows the input values to be normalized simultaneously. Thus, it facilitates the training process of the model by processing the training process of the model with normalized values (Amin et al., 2020). The Rectified Linear Unit (ReLU) activation function produces a linear positive output if the incoming values are positive, otherwise it produces the zero output for other values. It is usually used by default in most deep learning models (Eckle & Schmidt-Hieber, 2019).

Figure 3 Design showing the functioning of (A) convolutional block and (B) dense block used in the proposed model.

CBAM module

CBAM is a feedforward network module that examines the features obtained from the dataset by the model with the channel module and spatial module logic. The channel module enables the activation maps obtained from the layers to be directed to important regions (efficient features) on the map. The channel module combines and compresses the activation maps obtained from the layers. Thus, it reduces the cost of the model. The pooling layers used in the proposed model are of two types. These are average pooling and maximum pooling layers. The features extracted from these two pooling layers are processed by the spatial attention module and transferred to the convolutional layer. As a result, the feature information obtained from the pooling layers are combined in an accessible network and classifies these features among themselves with the MLP method. MLP acted as a feature identifier in this step. These features are transferred to Residual blocks in the proposed model (Toğaçar et al., 2019). The general design of the CBAM module and preferred CBAM parameter values in this study are shown in Fig. 4.

Figure 4 (A) Channel Attention Module and (B) Spatial Attention Module used in the proposed model.

The CBAM module consists of two modules called CAM (channel attention module) and SAM (spatial attention module), as shown in Fig. 4. It provides to deal with image parts related to the model from image data related to attention modules. The CAM module gives what to focus on whereas the SAM module gives us where to focus (Toğaçar, Ergen & Cömert, 2020b).

Residual block

One layer in standard neural networks directly transfers analysis results to the next layer. But the most important feature that distinguishes residual blocks from standard neural networks is that just as one layer will feed the next layer directly, two or three layers will feed the next layer directly. Here, residual blocks perform jumping between layers in the logic of operation. The layers to be skipped here are those that contribute little to the performance of the model. Thus, residual blocks provide both performance contribution and time savings to the model. In standard convolutional models, when the number of consecutive layers increases, the depth of the model increases and consequently decreases the performance of the optimization methods used in the model. Layers deemed unnecessary with residual blocks contribute directly to the training of the model without any negativity in the optimization methods (Yue, Fu & Liang, 2018). The general structure of the residual blocks used in this study is shown in Fig. 5.

Figure 5 (A) Accuracy graphs of the training and validation of the proposed model. (B) Loss graphs of the training and validation of the proposed model. (C) Confusion matrix.

Hypercolumn technique

Hypercolumn is a technique that performs the classification on pixels using hyper column. That is, each image given as an input to the model has a hyper column vector. These hyper vectors hold all the activation features of that pixel in the convolutional model. Thus, instead of deciding according to the pixel value in the final layer of the convolutional model in the classification process, it chooses the most efficient one by examining all the features in the hyper column vector. Thus, with this technique, the spatial location information of the most efficient feature is brought from the previous layers, and contributing to the classification process.

Basically, the essence of the Hypercolumn technique is based on heat maps. After the convolution layers of the model, this technique uses bilinear interpolation and creates a transition feature value using two feature values with Bilinear interpolation. In other words, bilinear interpolation creates a smooth transition value between two feature values. In this way, feature maps extracted from other layers of the model are added and it is processed with the sigmoid function. Heat maps extracted from the model are then combined to produce possible output values. This joining is done by the “Concatenate” function in the hypercolumn technique. In addition, with the Upsampling2D function, it keeps the neighboring pixel values of a pixel and transfers it to the required places (Toğaçar, Ergen & Cömert, 2020a; Toğaçar, Ergen & Cömert, 2020b).

Results

The experimental study was performed on a workstation with Intel(R) Xeon(R) Gold 6132 CPU @2.60 GHz, 64.0 GB DDR4 RAM, and 24 GB graphic card. The simulation environment was Python.

The performance of the proposed model was measured considering the confusion matrix and several standard metrics such as accuracy (Acc), sensitivity (Se), specificity (Sp), and F-score, derived from it. The mathematical descriptions of these metrics can be found in Cömert & Kocamaz (2018).

In this study, we considered the diagnosis of computed OM both in four classes and in five classes. The proposed model was performed both with and without the other class, which includes a limited number of samples belonging to the ear ventilation tube, foreign bodies in the ear, otitis externa, pseudo-membranes, and tympanoskleros. The whole dataset was divided into two parts that were training and test sets with rate of 80% and 20%, respectively. Also, the data augmentation techniques were employed in some of the experimental setup to balance the distribution of the recordings over the classes. In this manner, we carried out four different experiment in this study.

In the first experiment, the dataset was addressed as it is. So, totally five classes and 956 otoscope images were taken into account. The raw otoscope images were directly input to the proposed model. The accuracy and loss graphs of the training and validation of the proposed model and the obtained confusion matrix are shown in Fig. 6.

Figure 6 (A) Accuracy graphs of the training and validation of the proposed model. (B) Loss graphs of the training and validation of the proposed model. (C) Confusion matrix.

The numbers of samples in each class for the test set are distributed as 107 (normal), 24 (AOM), 13 (CSOM), 28 (Earwax), and 20 (Other), respectively. The model ensured an accuracy of 82.81%, a sensitivity of 67.74%, a specificiy of 93.54%, and a F-score of 72.70%. As inferred from the confusion matrix, the model ensured satisfying classification performances for Normal, AOM, and Earwax samples whereas the samples belonging to the CSOM and other classes were not predicted well enough. These results indicated that the model should be improved.

In the second experiment, all samples in the other class were excluded from the experiment due to the limited number of samples and all remaining parameters were kept as in the first experiment. In other words, 535 normal, 119 AOM, 63 CSOM, and 140 Earwax otoscope samples were considered in this experiment. This time, the proposed model was tested in four classes. Figure 7 presents the training and loss graphs of the proposed model with the confusion matrix.

Figure 7 (A) Accuracy graphs of the training and validation of the proposed model. (B) Loss graphs of the training and validation of the proposed model. (C) Confusion matrix.

The accuracy, sensitivity, specificity and F-scores of the proposed model were 86.63%, 74.07%, 92.22%, and 80.01%, respectively. Overall performance improvement was observed in the proposed model.

All samples except belonging to the CSOM class were predicted properly by the proposed model. Considering the results, it was seen that the proposed model had difficulty in distinguishing the samples in the CSOM class. The basic reason for this situation suffered from a limited number of samples in the mentioned class. In other words, the unbalanced data distribution among the classes in the dataset led to this situation. To tackle this issue, we embedded the data augmentation technique into the proposed model in the third experiment. As in previous experiments, firstly the whole dataset was divided into two parts with the specified rates. The data augmentation techniques were performed on only the samples in the training set to prevent overfitting. Since the highest number of samples belongs to the normal class, the data augmentation was carried out considering the normal class. The samples in AOM, CSOM, Earwax and Other classes were augmented three, seven, three and four times, respectively. In this manner, the distribution of the recordings among the classes was almost equalized and more data was provided for the model training. The accuracy and loss graphs of the proposed model and the obtained confusion matrix are given in Fig. 8. It was observed that the number of samples used for the model training was increased, and the results were considerably improved. The accuracy, sensitivity, specificity and F-score of the model was obtained as 92.19%, 86.15%, 97.06%, and 87.31%, respectively. As a result, promising results were achieved.

Figure 8 The residual block diagram used in the proposed model.

In the last experiment, the mentioned steps were followed as in the third experiment without considering the other class. The most efficient results were obtained in this experiment with an accuracy of 98.26%, sensitivity of 97.68%, specificity of 99.30%, and F-score of 96.89%. Figure 9 shows the accuracy of loss graphs of the proposed model with confusion matrix.

Figure 9 (A) Accuracy graphs of the training and validation of the proposed model. (B) Loss graphs of the training and validation of the proposed model. (C) Confusion matrix.

In summary, the results of the all experiments are reported in Table 2. Thanks to the proposed model, a novel OM diagnostic model ensured with high accuracy.

Table 2 The results of the experiments.

Experiment	# of class	Data augmentation	Acc
(%)	Se
(%)	Sp
(%)	F-Score (%)	
1	5	No	82.81	67.74	93.54	72.70	
2	4	No	86.63	74.07	92.22	80.01	
3	5	Yes	92.19	86.15	97.06	97.31	
4	4	Yes	98.26	97.68	99.30	96.89	

Discussion

In the first part of this section, we present a comparison considering the performances of the proposed model and pretrained CNNs such as AlexNet, VGG-16, VGG-19, GoogLeNet, and ResNet-50. In comparison to pretrained CNNs and the proposed model, we used the open-access tympanic membrane dataset without considering the other class. In this manner, a totally of 857 otoscope images were taken into account and the augmentation approach was used only on the training set. The pretrained CNNs were trained using the transfer learning approach. The parameters and values pairs for transfer learning are given in Table 3. The obtained performance results of the pretrained CNNs are reported in Table 4. When the results are examined, it is clearly seen that the superiority of the proposed model compared to the pretrained CNNs.

Table 3 The transfer learning parameters with value for pretrained CNNs.

Parameters	Values	
Mini-batch size	24	
Max epoch	32	
Initial learning rate	0.0001	
Learning rate schedule	Piecewise	
Learn rate drop factor	0.1	
Learn rate drop period	16	

Table 4 The performance results of the pretrained CNNs on OM diagnosis.

Models	Acc (%)	Se (%)	Sp (%)	F-Score (%)	
AlexNet	81.40	75.05	91.58	75.31	
VGG-16	81.40	73.00	90.70	75.47	
VGG-19	84.88	76.45	92.58	78.61	
GoogLeNet	80.23	72.97	89.96	74.81	
ResNet-50	80.23	68.49	89.29	72.90	
The proposed model	98.26	97.68	99.30	96.89	

In the second part of this section, we present another comparison between the proposed model and the related studies. It is also important to be aware that one-to-one comparison is not possible due to the different datasets, methods and parameters used in the related studies. However, such a comparison is also necessary to technically understand the models proposed in the field.

It is seen that the first decision support systems developed for the diagnosis of OM generally have focused on the morphological features of the otoscope images (Kuruvilla et al., 2013; Mironica, Vertan & Gheorghe, 2011). For this purpose, digital image processing techniques such as local binary pattern (LBP) and gray-level co-occurrence matrix (GLCM) have been used. These systems have been supported using traditional machine learning algorithms (Basaran et al., 2019a). Moreover, in some related studies, it has been observed that segmentation methods have been recommended to focus on TM annulus (Başaran, Cömert & Çelik, 2020).

Depending on the advances in deep learning, CNNs have been adopted to the OM diagnosis task (Basaran et al., 2019a, 2019b; Başaran, Cömert & Çelik, 2020). In the training of the CNNs, both the transfer learning (Cha et al., 2019) and training from scratch (Lee, Choi & Chung, 2019) approaches have been performed. In addition, the end-to-end learning model provided by the deep learning approach has eliminated the feature extraction and selection processes, providing a high generalization performance for many areas. For this reason, CNN networks appear to be of great interest and preferred over traditional methods. In this study, we have proposed a novel and advanced CNN model equipped with the CBAM module, hypercolumn technique and residual blocks. Thanks to these advanced methods, the proposed model was found superior to the traditional as well as pretrained CNNs.

Another important issue is related to datasets. As seen in Table 5, the vast majority of the studies were carried out on private datasets. The use of open-access datasets is extremely important to achieve repeatable results and to compare new algorithms with existing ones. Experiments in this study were carried out on an open-access dataset. This will allow researchers to easily compare new models.

Table 5 Comparison of the related studies.

Methods	# of images	Description of the classes	Public dataset	Acc (%)	
Digital image processing techniques and neural networks (Mironica, Vertan & Gheorghe, 2011)	186	Normal (111), OM (75)	No	73.11	
Preprocessing, automated segmentation of tympanic membrane, OM vocabulary and grammar, decision tree (Kuruvilla et al., 2013)	181	AOM (63), OME (70), Normal (48),	No	89.90	
Color, geometric and texture features, Gabor filter, local binary pattern, histogram of gradients, and AdaBoost (Shie et al., 2014)	865	Normal, AOM, OME, COM, OM	No	88.06	
Gray-level co-occurrence matrix, artificial neural network (Basaran et al., 2019b)	223	Normal (154) and AOM (69)	Yes	76.14	
Deep learning, pretrained CNN AlexNet, Transfer learning (Basaran et al., 2019a)	598	Normal (535) and COM (63)	Yes	98.77	
Faster R-CNN and pretrained CNNs, Transfer learning (Başaran, Cömert & Çelik, 2020)	282	Normal (154) and Abnormal (128)	Yes	90.48	
CNN, class activation map (Lee, Choi & Chung, 2019)	1,818	TM left and right sides	No	97.90	
Pretrained CNNs, transfer learning, Inception-V3 and ResnNet-101, Ensemble classifier, 5-fold cross-validation (Cha et al., 2019)	10,544	Normal, tympanic perforation, Attic retraction, OM externa, OME, Cerumen impaction	No	93.67	
Our study, CNN, CBAM, residual block, hypercolumn technique	857	AOM (119), CSOM (63), Earwax (140), Normal (535)	Yes	94.49	

Conclusions

OM is one of the most common illnesses. One of the main data sources of ENT specialists for the diagnosis of OM is otoscope images. In this study, a new OM decision support system was proposed in order to shorten the diagnostic process, reduce the rate of misdiagnosis and contribute to a more objective assessment. Unlike existing well-known CNNs, the proposed model was supported by CBAM and residual blocks and the hypercolumn technique was integrated into the proposed model. Thanks to these advanced methods, both the training period was shortened and superior results were obtained compared to the existing machine learning as well as pretrained CNN models. As a result, the proposed model achieved an accuracy of 98.26% sensitivity of 97.68% and specificity of 99.30% for automatically separating normal, AOM, CSOM and Earwax TM samples. Moreover, experimental studies were carried out on an open-access TM dataset. This will give researchers a chance to compare new models. Consequently, the results of this study indicate that CNN networks can produce efficient results for OM diagnosis. It has also been verified that new advanced methods integrated into the CNN architecture can increase the efficiency of OM diagnosis.

We would like to thank Mesut Toğaçar for his great contribution in data analysis and modeling in this paper.

Additional Information and Declarations

Competing Interests

Author Contributions

Data Availability

The authors declare that they have no competing interests.

Adi Alhudhaif conceived and designed the experiments, performed the experiments, analyzed the data, performed the computation work, authored or reviewed drafts of the paper, and approved the final draft.

Zafer Cömert conceived and designed the experiments, performed the computation work, prepared figures and/or tables, authored or reviewed drafts of the paper, and approved the final draft.

Kemal Polat performed the experiments, performed the computation work, authored or reviewed drafts of the paper, and approved the final draft.

The following information was supplied regarding data availability:

Source code is available at GitHub:

https://github.com/zcomert/omnet.

Data are available at Figshare:

POLAT, KEMAL (2021): eardrum.zip. figshare. Dataset. DOI 10.6084/m9.figshare.13648166.v1.

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
