# Peer review of "Otitis media detection using tympanic membrane images with a novel multi-class machine learning algorithm"

_PeerJ Computer Science, doi:10.7717/peerj-cs.405_

## Round 0.1 · original submission · Minor Revisions

Two reviewers have consistent recommendations. As a result, a revision is suggested.

·

Basic reporting

In this paper a novel convolutional neural network is proposed for determination of abnormalities or infections in human tympanic membrane using image processing methods. The paper technically seems useful regarding experimental applications. The proposed study aims to classify middle ear otoscope images based on the deep convolution neural networks (CNNs). To this aim, a novel deep CNN model including a combination of the channel and spatial model (CBAM), residual blocks, and hypercolumn technique have been embedded into the proposed model. The achieved results have been evaluated as promising.

Experimental design

To improve the quality of study, I have presented my comment as follows:
1. According to my knowledge, earwax is not an OM type, please clarity this issue on the manuscript.
2. There is a need for a more detailed explanation of some of the details of the Eardrum database in particular.
3. Authors must explain clearly the advantage of proposed algorithm.
4. In related work, paper with reference number [19] author mentioned the number of image sample as 10.544. How is it possible?

Validity of the findings

5. What is the initial learning rate vale of proposed model for training the dataset?
6. How many Positive and negative images taken for performing the tests of proposed model?
7. Author mentioned that training period shortened for the proposed work when compared with pre-trained CNNs, as we know transfer-learning takes less amount of time when compared to learning from basic on the particular dataset.

Additional comments

After minor revision, I think the manuscript can be proposed for publishing.
With regards.

Reviewer 2 ·

Basic reporting

The authors use methods that seem to be familiar to those in computer science, which I am not. Similarly, I am not the intended audience and could not understand the terminology.

As an example: the "down" function is defined in terms of the "down" function and it was not any clearer after reading the material. So I assume most readers are already familiar with this.

The emphasis on computational efficiency is important but does not appear useful in a data set with only few hundred observations. Otherwise this is OK.

Experimental design

They compare several clustering methods. This is appropriate.

Validity of the findings

The findings appear valid.

What is the "gold standard" for evaluating sensitivity and specificity? The article begins by saying the visual observation is "subjective and error-prone" so there is no way of telling whether their methods are correctly classifying OM cases.

Additional comments

Nothing additional

---

## Round 0.2 · accepted · Accept

The paper has been revised and improved. As a result, the paper is ready to be accepted.